# Disciplinary removal patterns among students with other health impairments and emotional disturbance: A three-year descriptive analysis of IDEA Part B data

**Melissa Beck Wells** [ID] *

Center for Teaching and Learning, SUNY Empire State College, Staten Island, New York, United States of America

* melissa.wells@sunyempire.edu

## Abstract

Students classified under Other Health Impairment (OHI), a category that includes Attention-Deficit/Hyperactivity Disorder (ADHD), are frequently disciplined in school settings despite ADHD being classified as a neurodevelopmental rather than a behavioral disorder. This study examined patterns of disciplinary removal among students served under the Individuals with Disabilities Education Act (IDEA), Part B, focusing on comparisons between OHI and Emotional Disturbance (ED). Using publicly available IDEA Part B discipline data, a descriptive longitudinal analysis was conducted across three academic years (2019–2020, 2020–2021, and 2021–2022). Percentage-based indicators were analyzed to examine the proportion of students ages 3–21 subject to disciplinary removal by disability category and state. Across all three years, students classified under OHI experienced disciplinary removal at rates comparable to, and in many states exceeding, those of students classified under ED. These patterns persisted despite year-to-year variation associated with the COVID-19 pandemic. Findings are interpreted through IDEA procedural safeguards and DSM-5 diagnostic frameworks, highlighting a potential misalignment between neurodevelopmental disability classification and behavioral support provision. The study discusses how Universal Design for Learning (UDL) may reduce disciplinary risk through proactive design, while emphasizing that individualized behavioral interventions, including Functional Behavioral Assessments and Behavior Intervention Plans, remain legally required when behavior is a manifestation of disability.

## Introduction

School discipline remains a persistent concern for students with disabilities, particularly when disability-related characteristics manifest as observable behaviors within classroom settings. Under the Individuals with Disabilities Education Act (IDEA),

**Data availability statement:** All data used in this study are publicly available through the U.S. Department of Education IDEA Section 618 Data Collection at https://data.ed.gov. This research data is available at https://osf.io/fxrd5.

**Funding:** The author(s) received no specific funding for this work.

**Competing interests:** The authors have declared that no competing interests exist.

students may be classified under a range of disability categories, each carrying implicit assumptions about academic and behavioral needs [1,2]. Emotional Disturbance (ED) is traditionally viewed as the disability category most closely associated with behavioral challenges and disciplinary intervention. However, emerging discipline data suggest that students classified under Other Health Impairment (OHI)—a category that includes Attention-Deficit/Hyperactivity Disorder (ADHD)—may experience comparable or higher exposure to disciplinary removal.

ADHD is not classified as a behavioral disorder in clinical diagnostic systems. The *Diagnostic and Statistical Manual of Mental Disorders, Fifth Edition, Text Revision* (DSM-5-TR) classifies ADHD as a neurodevelopmental disorder characterized by "impairing levels of inattention, disorganization, and/or hyperactivity-impulsivity" that interfere with functioning or development [5]. In school environments, these characteristics frequently manifest as impulsivity, difficulty sustaining attention, emotional dysregulation, and challenges with self-regulation—behaviors that are often interpreted as noncompliance rather than manifestations of disability.

IDEA discipline procedures establish explicit connections between disciplinary removal and the requirement to address underlying behavioral needs. Federal regulations require IEP teams to consider positive behavioral interventions when a child's behavior impedes learning [3] and to conduct Functional Behavioral Assessments (FBAs) and implement or revise Behavior Intervention Plans (BIPs) following removals that constitute a change of placement [4]. However, IDEA's federal reporting systems do not include direct indicators of FBAs or BIPs, limiting the ability to examine behavioral intervention patterns at scale.

The purpose of this study was to examine longitudinal patterns of disciplinary removal among students classified under OHI and ED using IDEA Part B discipline data across three academic years and to interpret these patterns through federal policy, DSM-5 diagnostic frameworks, and peer-reviewed Universal Design for Learning (UDL) research.

## Methods

### Data source

This study used publicly available administrative data from the U.S. Department of Education's IDEA Section 618 Data Collection. Specifically, data were drawn from the IDEA Part B discipline table reporting the percentage of children and students ages 3–21 subject to disciplinary removal by disability category and state.

### Study design

A descriptive longitudinal design was employed examining three academic years: 2019–2020, 2020–2021, and 2021–2022. Percentage-based indicators were selected to facilitate cross-state and cross-year comparisons and to avoid double-counting students across disciplinary events.

## Measures

### Disability categories

Analyses focused on students classified under Other Health Impairment and Emotional Disturbance. ADHD is reported federally within the OHI category and is not disaggregated separately in IDEA discipline data.

### Outcome variable

The primary outcome was the percentage of students subject to at least one disciplinary removal during the academic year.

### Analytic approach

Data were analyzed descriptively. Because national weighted estimates are not provided in the IDEA discipline tables, median state-level percentages were calculated to summarize disciplinary removal patterns while minimizing the influence of outlier states. No inferential statistical testing was conducted, consistent with the study's focus on identifying system-level patterns rather than causal effects.

### Ethics statement

This study used publicly available, de-identified secondary data and did not involve human subjects as defined under federal regulations. Institutional Review Board review was not required.

## Results

Across all three academic years examined, students classified under Other Health Impairment experienced disciplinary removal at rates comparable to or exceeding those of students classified under Emotional Disturbance (Table 1).

Median state-level disciplinary removal percentages for OHI ranged from 26.06% to 28.13% across years, while median percentages for ED ranged from 11.00% to 12.52% (Table 1). This pattern remained stable across years despite fluctuations associated with the COVID-19 pandemic.

State-level analyses revealed substantial variability; however, in a majority of states, students classified under OHI were disciplined at rates comparable to or higher than students classified under ED in each year examined, indicating a persistent system-level pattern rather than a single-year anomaly. Trends in median disciplinary removal rates across years are illustrated in Fig 1.

## Discussion

### IDEA discipline as a trigger for behavioral intervention

IDEA regulations explicitly link disciplinary removal to the requirement to address underlying behavioral needs. Federal law requires IEP teams to consider behavioral supports when "a child's behavior impedes the child's learning or that of

**Table 1. Median percentage of students subject to disciplinary removal by disability category and year.**

| Year | OHI (%) | ED (%) |
|---|---|---|
| 2019–2020 | 26.06 | 12.52 |
| 2020–2021 | 28.13 | 11.87 |
| 2021–2022 | 27.30 | 11.00 |

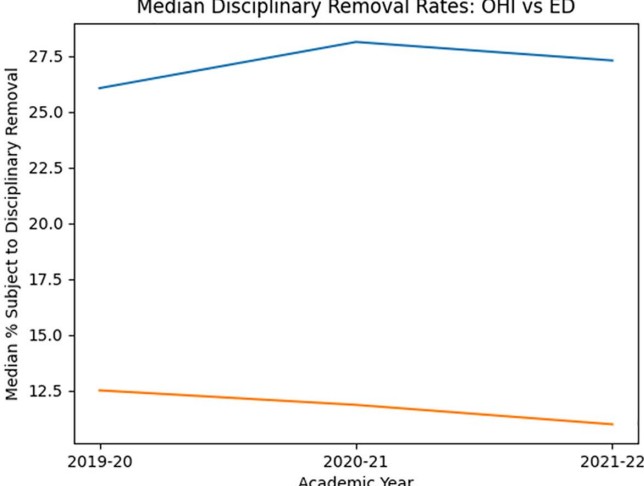

**Fig 1. Median percentage of students subject to disciplinary removal for Other Health Impairment (OHI) and Emotional Disturbance (ED) across three academic years (2019–2020 to 2021–2022).**

others" [3]. When disciplinary removals constitute a change of placement, schools must "conduct a functional behavioral assessment, and implement a behavior intervention plan, or, if a behavior intervention plan has already been developed, review and modify it" [4]. Disciplinary removal therefore functions not only as a consequence but as a procedural trigger for formal behavioral assessment and intervention under IDEA.

## DSM-5 context and diagnostic misalignment

The DSM-5-TR classifies ADHD as a neurodevelopmental disorder rather than a behavioral disorder, emphasizing impairments in attention, impulsivity, and self-regulation that manifest across settings, including schools [5]. These characteristics frequently present as observable behaviors that conflict with classroom norms, increasing the likelihood of disciplinary responses even in the absence of emotional disturbance or intentional misconduct.

Students identified under Emotional Disturbance are more likely to have behavioral needs explicitly recognized at eligibility, often resulting in earlier implementation of Behavior Intervention Plans and therapeutic supports. The comparable or higher disciplinary removal rates observed for students classified under OHI suggest a misalignment between diagnostic classification and educational response, with ADHD-related behaviors more likely to be addressed reactively through discipline rather than proactively through behavioral intervention.

## Universal design for learning as prevention, not replacement

A growing body of peer-reviewed research supports Universal Design for Learning as an evidence-aligned framework for reducing barriers that disproportionately affect students with neurodevelopmental disabilities, including ADHD [6–10]. UDL emphasizes proactive instructional design across three principles—multiple means of engagement, representation, and action/expression—intended to anticipate learner variability rather than respond reactively to behavioral challenges.

Recent studies indicate that UDL-aligned practices are associated with improvements in student engagement, self-regulation, and behavioral persistence, particularly for learners with executive functioning and attention-related challenges [6–9]. In K–12 contexts, predictable routines, flexible pacing, and explicit self-regulation supports have been shown to reduce behavioral escalation and disciplinary risk [7,10].

However, UDL does not replace IDEA's legal requirements for individualized behavioral intervention. When behavior persists or escalates as a manifestation of disability, IDEA mandates a shift from universal prevention to individualized, data-driven intervention through FBAs and BIPs [3,4]. The persistence of elevated disciplinary removal rates for students classified under OHI across three academic years suggests that UDL may be underutilized or inconsistently implemented and that ADHD-related behavioral needs may not be systematically addressed until disciplinary thresholds are reached.

## Limitations

This study relied on publicly available administrative data and did not include IEP-level indicators such as the presence of FBAs or BIPs. ADHD could not be disaggregated from the OHI category. Findings are descriptive and do not establish causal relationships.

## Conclusion

Findings challenge assumptions that Emotional Disturbance is the disability category most associated with school discipline. Students with ADHD, classified under Other Health Impairment, experience disciplinary removal at consistently high rates across multiple years. Addressing these patterns requires both preventative universal design strategies and rigorous adherence to IDEA-mandated individualized behavioral interventions when behavior is a manifestation of disability.

## Author contributions

**Conceptualization:** Melissa Beck Wells.

**Data curation:** Melissa Beck Wells.

**Formal analysis:** Melissa Beck Wells.

**Funding acquisition:** Melissa Beck Wells.

**Investigation:** Melissa Beck Wells.

**Methodology:** Melissa Beck Wells.

**Project administration:** Melissa Beck Wells.

**Resources:** Melissa Beck Wells.

**Software:** Melissa Beck Wells.

**Supervision:** Melissa Beck Wells.

**Validation:** Melissa Beck Wells.

**Visualization:** Melissa Beck Wells.

**Writing – original draft:** Melissa Beck Wells.

**Writing – review & editing:** Melissa Beck Wells.

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
