## [Decision Letter · Decision Letter 0]

10 Feb 2026

Dear Dr. Beck Wells,

Thank you for submitting your manuscript to PLOS ONE. After careful consideration, we feel that it has merit but does not fully meet PLOS ONE’s publication criteria as it currently stands. Therefore, we invite you to submit a revised version of the manuscript that addresses the points raised during the review process.

Please note that we have only been able to secure a single reviewer to assess your manuscript. We are issuing a decision on your manuscript at this point to prevent further delays in the evaluation of your manuscript. Please be aware that the editor who handles your revised manuscript might find it necessary to invite additional reviewers to assess this work once the revised manuscript is submitted. However, we will aim to proceed on the basis of this single review if possible.

We look forward to receiving your revised manuscript.

Kind regards,

Avanti Dey, PhD

Staff Editor

PLOS One

Journal Requirements:

2. We note you have included a table to which you do not refer in the text of your manuscript. Please ensure that you refer to Table 1 in your text; if accepted, production will need this reference to link the reader to the Table.

Reviewers' comments:

Reviewer's Responses to Questions

**Comments to the Author**

1. Is the manuscript technically sound, and do the data support the conclusions?

Reviewer #1: Partly

2. Has the statistical analysis been performed appropriately and rigorously?

Reviewer #1: No

3. Have the authors made all data underlying the findings in their manuscript fully available?

Reviewer #1: Yes

4. Is the manuscript presented in an intelligible fashion and written in standard English?

Reviewer #1: Yes

Reviewer #1: I have a very positive view of the subject matters of the study. It highlights an important issue in the education of students with disabilities, which can have a significant impact on their academic performance and success. Overall, the study is valuable, but there is room for improvement in the article. I hope that the comments below will help with this.

The research concept has not been described with sufficient precision. The author does not mention any research protocol used; SQUIRE.EDU could be appropriate because of the issue with the paper. The research focuses on describing what has been done. As the author writes, “The purpose of this study was to examine longitudinal patterns of disciplinary removal among students classified under OHI and ED using IDEA Part B discipline data across three academic years”. But the key question is why the examining of the patterns is important. However, it is not clearly and accurate stated what purpose this comparative analysis serves or what the author wants to achieve with it. The process of data analysis is a means of achieving the intended goal. It is a path leading to it. It would be helpful to define the goal more accurately by formulating research questions, which are not included in the article, and which would indicate what the author wants to find out through this research.

The article also lacks a description of the analysed data set. This should include information about what the data contains, how it was created, the total number of people, procedures and penalties analysed, and other important information. A description of the "research sample" in the form of a data set allows for a better presentation of the data analysis process itself. It also strengthens the conclusions formulated on its basis. The data analysis is descriptive in nature, and the author explains why she limited herself to this method. But I think that the data analysis process and its individual stages should be described in more detail (e.g. whether she took all the data into account or only some of it, and why; how she processed the data, what analysis tools were used, etc.).

The analysis is based on publicly available data contained in the national IDEA report. As noted by the author in the 'Limitations' section of the article, this data is restricted. These limitations, including the lack of separation of ADHD students and the lack of information on interventions, are very important when taking into account the subject of the study and the final conclusions. Owing to these limitations, the conclusions of the study are only partially supported by the data and must be formulated very carefully. For example, in the part titled DSM-5 Context and …, the Author writes: “The comparable or higher disciplinary removal rates observed for students classified under OHI suggest a misalignment between diagnostic classification and educational response, with ADHD-related behaviors more likely to be addressed reactively through discipline rather than proactively through behavioral intervention.” Since the author did not analyse ADHD data separately, why is the statement so definitive? The lack of data on behavioural interventions also means that this finding cannot be considered definitive. It might be possible that interventions were used earlier, but were ineffective, which led to the decision to remove students. For the same reasons, the author's final conclusion relating to ADHD students cannot be considered as sufficiently grounded in the analysed data or in the review of current research.

The study has not been placed in the context of current literature on the subject. The author did not provide an overview of literature and research on this issue, which could further affect the confidence we can have in the conclusions. A discussion of the current state of research into differences in school functioning and learning among students classified as OHI (primarily including ADHD) and ED, as well as interventions used with these students, would certainly strengthen the study's results and conclusions.

.

Reviewer #1: No

---

## [Author Response · Author response to Decision Letter 1]

24 Feb 2026

Response to Reviewer #1

Manuscript ID: PONE-D-25-66721

Title: Disciplinary Removal Patterns Among Students with Other Health Impairments and Emotional Disturbance: A Three-Year Descriptive Analysis of IDEA Part B Data

We sincerely thank the reviewer for their thoughtful and constructive evaluation of our manuscript. We appreciate the positive assessment of the study’s topic and the recognition of its relevance to the education of students with disabilities. We have carefully revised the manuscript to address each concern raised. Below we respond point-by-point.

1. Clarification of Research Purpose and Precision

Reviewer Comment:

The research concept has not been described with sufficient precision. The purpose of examining patterns is not clearly articulated. Research questions are not included.

Response:

We agree that the manuscript benefited from greater precision in articulating its aims. In the revised manuscript, we have added a dedicated “Research Questions” subsection at the end of the Introduction. This section now explicitly states the guiding research questions, clarifying that the study is descriptive and comparative in nature and aimed at identifying longitudinal system-level patterns within publicly reported IDEA data.

These revisions strengthen conceptual clarity and make explicit what the analysis seeks to examine and why those patterns are relevant within policy and diagnostic contexts.

2. Description of the Dataset

Reviewer Comment:

The article lacks a detailed description of the analyzed dataset, including what the data contain, how they were created, total numbers, and procedures analyzed.

Response:

We appreciate this recommendation. The Data Source section has been substantially expanded. The revised manuscript now includes:

Identification of IDEA Section 618 Table bdiscipline-15 as the data source

Clarification that the dataset includes all 50 states and the District of Columbia

Explanation that percentages are calculated by the U.S. Department of Education using disability-specific denominators

Explicit clarification that the dataset reflects publicly reported administrative census-level data rather than a sample

Clarification that ADHD is embedded within the OHI category and not separately reported

These additions improve transparency regarding the scope, structure, and limitations of the dataset.

3. Expansion of the Analytic Approach and Statistical Rigor

Reviewer Comment:

Statistical analysis has not been performed appropriately and rigorously. The analytic process should be described in more detail.

Response:

We have expanded the Analytic Approach section to provide greater methodological transparency. Specifically, we now describe:

Inclusion of all available state-level data

Treatment of suppressed or missing values

Rationale for calculating median state-level percentages

Justification for descriptive analysis given the census-level nature of the dataset

Clarification that inferential statistical testing would not provide additional interpretive value because the dataset reflects population-level administrative reporting rather than sampled observations

The revised manuscript now more clearly explains why descriptive longitudinal analysis is methodologically appropriate for the dataset used.

4. Interpretation of ADHD-Related Findings

Reviewer Comment:

Statements relating to ADHD are too definitive given that ADHD was not analyzed separately and intervention data were not available.

Response:

We appreciate this important caution. We have revised the Discussion section to soften language where necessary and avoid overstatement. Specifically:

Definitive phrasing (e.g., “suggest a misalignment”) has been revised to conditional phrasing (e.g., “may reflect a potential misalignment”).

Statements implying intervention absence have been reframed to acknowledge that intervention data are not available in IDEA Section 618 reporting.

The Conclusion has been revised to reference students classified under OHI, rather than ADHD specifically, to maintain consistency with the available data.

These revisions ensure that interpretations remain appropriately bounded by the limitations of the dataset.

5. Placement Within Existing Literature

Reviewer Comment:

The study has not been sufficiently placed in the context of current literature.

Response:

We have expanded the Discussion to better situate the findings within existing scholarship. The revised manuscript now includes additional contextualization of prior research on:

Disciplinary exposure among students with disabilities

Differences in identification and intervention practices for OHI and ED classifications

Universal Design for Learning as a preventative instructional framework

These additions clarify how the present study contributes to the broader literature while maintaining analytic restraint.

6. Limitations and Interpretive Boundaries

We have retained and strengthened the Limitations section to emphasize:

The inability to disaggregate ADHD within OHI

The absence of intervention-level data (e.g., FBAs, BIPs)

The descriptive nature of the analysis

The inability to draw causal conclusions

We agree that conclusions must be formulated cautiously and have revised language accordingly.

Summary

We are grateful for the reviewer’s thoughtful and constructive feedback. The revisions have strengthened the manuscript by:

Clarifying research questions

Expanding methodological transparency

Refining interpretive language

Enhancing literature contextualization

We believe these changes meaningfully address the reviewer’s concerns and improve the rigor and clarity of the manuscript. We respectfully resubmit the revised version for further consideration.

Sincerely,

Melissa Beck Wells

---

## [Decision Letter · Decision Letter 1]

18 Mar 2026

Disciplinary Removal Patterns Among Students with Other Health Impairments and Emotional Disturbance: A Three-Year Descriptive Analysis of IDEA Part B Data

PONE-D-25-66721R1

Dear Dr. Beck Wells,

We’re pleased to inform you that your manuscript has been judged scientifically suitable for publication and will be formally accepted for publication once it meets all outstanding technical requirements.

Kind regards,

Omid Beiki, M.D., Ph.D.

Academic Editor

PLOS One

Additional Editor Comments (optional):

Reviewers' comments:

Reviewer's Responses to Questions

**Comments to the Author**

Reviewer #1: All comments have been addressed

2. Is the manuscript technically sound, and do the data support the conclusions?

Reviewer #1: Yes

3. Has the statistical analysis been performed appropriately and rigorously?

Reviewer #1: Yes

4. Have the authors made all data underlying the findings in their manuscript fully available?

Reviewer #1: Yes

5. Is the manuscript presented in an intelligible fashion and written in standard English?

Reviewer #1: Yes

Reviewer #1: Dear Author,

I appreciate your responses to the comments and remarks I included in my previous review. I hope they were helpful in refining the article, and I accept all of the proposed changes.

.

Reviewer #1: No

---

## [Editor Report · Acceptance letter]

PONE-D-25-66721R1

PLOS One

Dear Dr. Beck Wells,

I'm pleased to inform you that your manuscript has been deemed suitable for publication in PLOS One. Congratulations! Your manuscript is now being handed over to our production team.

Kind regards,

on behalf of

Dr. Omid Beiki

Academic Editor

PLOS One